

# Effects of dietary supplements on bone turnover markers in women after menopause: a network meta-analysis

Yan Wei[1,2], Congjie Lei[3], Yue Zhong[2] and Hongchun Shen[1,4]

[1] Department of Nephropathy, The Affiliated Traditional Chinese Medicine Hospital, Southwest Medical University, Luzhou, Sichuan, China
[2] Nutritional Department, Zigong Hospital of TCM, Zigong, Sichuan, China
[3] Zigong Hospital of TCM, Zigong, Sichuan, China
[4] College of Integrated Chinese and Western Medicine, Southwest Medical University, Luzhou, Sichuan, China

## ABSTRACT

**Background**. Diminished estrogen levels in women after menopause contribute to an elevated risk of decreased bone mineral density (BMD) and disturbed bone metabolism. Dietary supplements are extensively employed as substitutes for prescription drugs, serving as a significant approach to modulate bone metabolism and improve bone health. Nevertheless, a lack of robust evidence prevents clinicians and patients from making the best-informed choices at present. Accordingly, a network meta-analysis (NMA) was performed to provide a comprehensive comparison of the effects of different dietary supplements on bone turnover biomarkers among postmenopausal women.

**Methods**. PubMed, Embase, Cochrane Library, and Web of Science databases were retrieved from their inception to November 20, 2024. Randomized controlled trials (RCTs) assessing the strength of dietary supplements in women after menopause were adopted in this research. Primary outcome indicators encompassed C-terminal telopeptide of type I collagen (CTX), procollagen type I N-terminal propeptide (P1NP), osteocalcin (OC), bone alkaline phosphatase (BAP), alkaline phosphatase (ALP), and N-terminal telopeptide of type I collagen (NTX). Risk of bias assessment was performed for each enrolled trial utilizing the ROB 2.0. A Bayesian NMA was conducted utilizing the R software (V4.4.1). Publication bias was assessed utilizing Stata 15.1.

**Results**. Forty-three RCTs involving 5,184 postmenopausal women were adopted. Research findings revealed that CTX levels decreased most substantially with vitamin E supplementation (SUCRA: 82.37%). Combining vitamin D and vitamin K most effectively elevated P1NP (SUCRA: 100.00%) and OC (SUCRA: 97.05%) levels. The greatest increase in BAP levels (SUCRA: 95.50%) was observed with vitamin K supplementation. Calcium supplementation yielded the largest elevation in ALP levels (SUCRA: 96.68%). Furthermore, protein supplementation resulted in the most significant reduction in NTX levels (SUCRA: 81.20%).

**Conclusion**. Dietary supplementation may serve as an important strategy for enhancing bone health in women after menopause by regulating bone metabolism. By promoting bone formation and inhibiting bone resorption, vitamin D, vitamin E, vitamin K, calcium, and protein may help mitigate the progression of osteoporosis.

Corresponding author
Hongchun Shen,
Shenhongchun79@163.com

# INTRODUCTION

Osteoporosis poses a substantial threat to the health and well-being of older adults, presenting a serious challenge to public health. Its global prevalence is approximately 18.3%, with about 30% of postmenopausal women affected (*Vilaca, Eastell & Schini, 2022*). In the postmenopausal period, a precipitous drop in estrogen levels causes decreased bone mineral density (BMD) and altered bone metabolism among women. These changes elevate the likelihood of osteoporosis and fractures (*Azizieh et al., 2019*). Research indicates that women with osteoporosis have a threefold elevated risk of fracture in comparison to women with normal bone density (*Vallibhakara et al., 2021*). In 2019, the European Union recorded 4.28 million cases of fractures attributable to osteoporosis annually. By 2034, the number is forecast to climb to 5.34 million (*Kanis et al., 2021*). Among women aged 50 and above, the WHO estimates that approximately 21.2% of them are affected by osteoporosis. Worldwide, up to 70 fragility fractures occur each minute in women aged 55 and over (*Nunkoo et al., 2024*).

Osteoporosis is a major contributor to the incidence and mortality rates among postmenopausal women, underscoring the crucial need for preventive interventions in this population. The occurrence of osteoporotic fractures is closely correlated with abnormal bone metabolic markers. For instance, bone alkaline phosphatase (BAP), a significant bone metabolic marker, is typically elevated in association with increased bone formation. Numerous studies have demonstrated a positive correlation between BAP and bone mineral density (BMD) (*Du, Chen & Shui, 2024*). In patients receiving treatment for osteoporotic fractures, a reduction in BAP levels generally predicts improved bone metabolism and increased bone density (*Zhu et al., 2021*). However, in postmenopausal women, the regulatory effects and clinical significance of dietary supplements on bone metabolic markers require further investigation. At present, a variety of medications are recognized for their beneficial impact on bone metabolism; nevertheless, their clinical application is limited by negative effects, such as abnormal reactions in hormone-sensitive organs, gastrointestinal problems, atypical fractures, and pain at the injection site (*Han et al., 2022*). As a result, dietary supplements are increasingly applied as alternatives to prescription drugs for regulating bone metabolism and improving bone health. Frequently utilized dietary supplements encompass vitamin D, calcium, omega-3 fatty acids, probiotics, and botanical supplements (*De Sire et al., 2022*). These supplements have been shown to play a critical role in modulating bone metabolism in numerous basic and clinical studies. For instance, vitamin D and calcium are among the most commonly used bone health ingredients, jointly promoting calcium absorption and maintaining BMD (*Méndez-Sánchez et al., 2023*). Long-chain omega-3 polyunsaturated fatty acids, including eicosapentaenoic acid (EPA, 20:5n-3) and docosahexaenoic acid (DHA, 22:6n-3), found in fatty fish, have been demonstrated to inhibit osteoclast formation, decrease inflammatory

cytokines, enhance calcium absorption, and elevate bone calcium levels (*Ali et al., 2024*; *Sharma & Mandal, 2020*; *Shawl et al., 2024*). Emerging dietary supplements, probiotics and prebiotics, may exert a beneficial influence on BMD and bone metabolism by modulating the gut microbiota and enhancing intestinal health (*Zhang et al., 2023*). In addition, phytochemicals and polyphenols exhibit protective effects against bone loss by modulating antioxidant pathways, reducing inflammation, promoting osteoblast differentiation, inhibiting osteoclastogenesis, and regulating bone immune responses (*Hou, Zhang & Yang, 2019*; *Torre, 2017*; *Zhao et al., 2018*). Bone turnover markers, serving as dynamic indicators of bone metabolism, can reflect the activity of bone formation and resorption at an early stage. They are important biomarkers for evaluating the outcomes of dietary interventions. The use of bone formation markers such as procollagen type 1 N-terminal propeptide (P1NP) and bone resorption markers such as C-terminal telopeptide of type I collagen (CTX) enables the dynamic monitoring of the fine-tuned effects of dietary supplements on bone turnover processes, thereby elucidating their mechanisms of action. For instance, supplementation with calcium and vitamin D can lower CTX levels, reflecting the inhibition of bone resorption activity (*Méndez-Sánchez et al., 2023*). Omega-3 fatty acids reduce CTX levels by mitigating inflammatory responses and concurrently enhance P1NP levels, illustrating their role in promoting bone formation (*Ali et al., 2024*; *Sharma & Mandal, 2020*). Nonetheless, the difference in the effects of various dietary supplements and their potential interactions need to be validated in more extensive studies.

Network meta-analysis (NMA) is a statistical approach for integrating data from multiple studies. In contrast to traditional meta-analyses that only compare two directly competing interventions, NMA connects multiple interventions evaluated in different studies *via* "common comparators" into an integrated network, allowing for the simultaneous evaluation of the relative effects of all treatment options. Current studies are often limited to comparing two or a few treatment approaches, which hinders a comprehensive analysis of the relative effectiveness across all available therapeutic options. Hence, it is imperative to conduct integrative research that concurrently evaluates the effects of multiple treatment options. Consequently, a network meta-analysis (NMA) methodology was employed to systematically assess the effects of different dietary supplements on bone turnover markers (BTMs) among females after menopause, encompassing vitamins, calcium, omega-3 fatty acids, probiotics, and botanical supplements, thus providing a scientific basis for clinical practice and offering insights for future research directions.

Existing studies are often limited to comparing only two or three interventions, making it challenging to comprehensively assess the relative effectiveness of various therapies. Therefore, this study employed NMA to systematically evaluate the effects of dietary supplements, including vitamins, calcium supplements, omega-3 fatty acids, probiotics, and botanicals, on BTMs in postmenopausal women. The findings will provide a scientific basis for clinical practice and offer insights for future research directions.

## METHODS

This NMA adhered to a pre-set study protocol, which was registered in PROSPERO (CRD42024626695). This meta-analysis was performed in compliance with the Preferred

Reporting Items for Systematic Reviews and Meta-Analyses (PRISMA) Statement 2020 (*Page et al., 2021*).

## Data source and acquisition

All English-language studies were retrieved in the PubMed, Embase, Cochrane Library, and Web of Science databases, spanning from their inception to November 20, 2024. The retrieval was carried out by combining subject headings and text words, and the medical subject headings were as follows: "Dietary Supplements", "Probiotics", "Fatty Acids, Omega-3", "Vitamins", "Postmenopause". Details of the search strategy are available in Table S1. To ensure thorough retrieval, the references of existing systematic reviews were further searched.

## Inclusion and exclusion criteria
### Inclusion criteria
Studies were eligible for inclusion if they met the following conditions: (1) Studies enrolled postmenopausal women; (2) Studies explicitly described two or more of the following interventions: placebo, protein, probiotics, prebiotics, polyphenols, calcium, lycopene, vitamin B, vitamin D, vitamin E, vitamin K, and omega-3; (3) The outcome measures included C-terminal telopeptide of type I collagen (CTX), procollagen type I N-terminal propeptide (P1NP), osteocalcin (OC), bone alkaline phosphatase (BAP), alkaline phosphatase (ALP), and N-terminal telopeptide of type I collagen (NTX); (4) Study type: all reports included in this study were randomized controlled trials (RCTs).

### Exclusion criteria
Studies were excluded if they were: (1) Animal or cell trials, case reports, scientific experimental schemes, comments, letters, editorials, and conference papers; (2) Articles lacking essential data; (3) Duplicates; (4) Studies lacking accessible full texts.

## Extraction of data

Retrieved articles were imported into EndNote 21. Two researchers separately assessed titles and abstracts for inclusion and exclusion, followed by a full-text review. Disagreements were addressed by discussion or consulting a third researcher. Literature screening begun on December 19, 2024. Two researchers separately applied Excel to collect relevant information from the selected studies, specifically the first author, year of publication, location, study type, sample size, age, intervention measures, follow-up duration, and outcome parameters (including CTX, P1NP, OC, BAP, ALP, and NTX). Data extraction begun on December 29, 2024.

## Quality assessment

Utilizing the Cochrane risk-of-bias (RoB 2.0), two separate authors evaluated the risk of bias in the selected RCTs (*Sterne et al., 2019*). This tool assessed five key domains: random sequence generation, allocation concealment, the application of blinding, missing data management, and selective reporting. Each domain was assessed as "high risk", "low risk", or "some concerns" accordingly. The quality of the included studies was independently evaluated by two authors, with a third author stepping in to settle any discrepancies.

## Statistical analysis

The outcome measures of all included studies were continuous variables. The weighted mean difference (WMD) was employed as the effect size. A Bayesian NMA model was constructed utilizing Markov chain Monte Carlo (MCMC) methods. This model was iterated to derive estimates of the relative effectiveness among various treatment strategies (*Shim et al., 2019*). Four Markov chains were run during model fitting, with a burn-in of 10,000 iterations to exclude samples obtained during the non-converged phase of the Markov chains. Each chain was iterated 50,000 times, with a logging interval of 10. Initial values for the Markov chains were set to 2.5. This procedure was conducted to estimate the posterior distributions. An NMA is valid under three fundamental assumptions: transitivity, homogeneity, and consistency. The mtc.anohe function from the GeMTC package was employed to assess heterogeneity in each direct comparison. According to the Cochrane standards (*Higgins et al., 2003*), an $I^2$ statistic below 50% typically indicates low to moderate heterogeneity, which is considered acceptable. Therefore, we employed an overall $I^2$ statistic below 50%. The mtc.nodesplit function within the GeMTC package was applied to explore the inconsistency among direct and indirect evidence utilizing the node-splitting method. A $p$-value > 0.05 was interpreted as no marked inconsistency, thus satisfying the consistency assumption. Convergence of study outcomes was evaluated through an inconsistency test. A network structure was generated, where interventions were nodes, and edges represented the standardized mean differences (SMDs) between interventions. This structure was assessed against a target convergence value of 1. League tables were created to visualize the relative effect of each intervention *vs.* placebo. An effect was considered statistically significant if the confidence interval for the pooled SMD excluded zero. Cumulative ranking plots were analyzed. Cumulative ranking probabilities were estimated for all interventions, and the surface under the cumulative ranking curve (SUCRA) was calculated and reported as the cumulative probability ranking. In a network diagram, interventions were represented by nodes, and direct comparisons between interventions were indicated by edges connecting those nodes. The size of each node corresponded to the total number of trials on each intervention, while the edge thickness reflected the number of trials informing the specific head-to-head comparison. To evaluate potential publication bias, a funnel plot was utilized. The presence of small-study effects and potential publication bias was assessed by examining the symmetry of graphical displays. Asymmetry in the graph suggests the potential for publication bias or other sources of heterogeneity between studies. All statistical analyses were conducted utilizing R (V4.4.1) and STATA (V15.1).

# RESULTS

## Literature screening

From databases, 38,718 articles were retrieved. *Via* EndNote, duplicates were deleted, yielding a reduced set of 10,047 articles. After titles and abstracts were reviewed, 28,576 obviously irrelevant articles were eliminated. The 95 articles meeting the initial screening criteria were subjected to further review. Study type, population, intervention, and outcome

measures were reassessed. Then, eight articles were removed due to missing full text, three due to mismatched interventions, 34 due to insufficient data, four duplicates, and three research protocols. Finally, 43 articles were included in the study (*Aloia et al., 2013*; *Aoe et al., 2005*; *Argyrou et al., 2020*; *Arjmandi et al., 2005*; *Atkinson et al., 2004*; *Braam et al., 2003*; *Bristow et al., 2014*; *Chen et al., 1997*; *Cúneo et al., 2010*; *Dong et al., 2014*; *Grados et al., 2003*; *Gregori et al., 2024*; *Han et al., 2022*; *Herrmann et al., 2007*; *Holloway et al., 2007*; *Jafarnejad et al., 2017*; *Koitaya et al., 2009*; *Koitaya et al., 2014*; *König et al., 2018*; *Macdonald et al., 2013*; *Mackinnon et al., 2011*; *Majidi et al., 2021*; *Maria et al., 2017*; *Meeta et al., 2022*; *Emaus et al., 2010*; *Nahas-Neto et al., 2018*; *Ooms et al., 1995*; *Panahande et al., 2019*; *Pérez-Alonso et al., 2024*; *Prickett, Howe & Espiner, 2023*; *Rajatanavin et al., 2013*; *Ruml et al., 1999*; *Schult et al., 2004*; *Shen et al., 2018*; *Stone et al., 2017*; *Tai et al., 2012*; *Takimoto et al., 2018*; *Ulrich et al., 2004*; *Vallibhakara et al., 2021*; *Vanitchanont et al., 2024*; *Vanlint & Ried, 2012*; *Wu et al., 2022*; *Ye et al., 2006*). Figure 1 illustrates the detailed screening process.

## Baseline features of the studies

Table 1 presents the basic features of the included studies. This analysis incorporated 43 RCTs (*Aloia et al., 2013*; *Aoe et al., 2005*; *Argyrou et al., 2020*; *Arjmandi et al., 2005*; *Atkinson et al., 2004*; *Braam et al., 2003*; *Bristow et al., 2014*; *Chen et al., 1997*; *Cúneo et al., 2010*; *Dong et al., 2014*; *Grados et al., 2003*; *Gregori et al., 2024*; *Han et al., 2022*; *Herrmann et al., 2007*; *Holloway et al., 2007*; *Jafarnejad et al., 2017*; *Koitaya et al., 2009*; *Koitaya et al., 2014*; *König et al., 2018*; *Macdonald et al., 2013*; *Mackinnon et al., 2011*; *Majidi et al., 2021*; *Maria et al., 2017*; *Meeta et al., 2022*; *Emaus et al., 2010*; *Nahas-Neto et al., 2018*; *Ooms et al., 1995*; *Panahande et al., 2019*; *Pérez-Alonso et al., 2024*; *Prickett, Howe & Espiner, 2023*; *Rajatanavin et al., 2013*; *Ruml et al., 1999*; *Schult et al., 2004*; *Shen et al., 2018*; *Stone et al., 2017*; *Tai et al., 2012*; *Takimoto et al., 2018*; *Ulrich et al., 2004*; *Vallibhakara et al., 2021*; *Vanitchanont et al., 2024*; *Vanlint & Ried, 2012*; *Wu et al., 2022*; *Ye et al., 2006*), comprising 5,184 females after menopause. The average age of the individuals varied between 49.7 and 80.6 years. These studies, published between 1995 and 2024, encompass data from 20 countries. Specifically, 10 investigations were carried out within the United States, five in Japan, three in Iran and Thailand respectively, and two each in Australia, China, South Korea, the Netherlands, New Zealand, and the United Kingdom. Furthermore, one study was each performed in Argentina, Canada, France, Germany, Greece, India, Norway, Spain, Sweden, and Brazil.

## Quality assessment

A risk of bias evaluation was completed for each of the 43 studies selected, by utilizing the ROB 2.0 tool. The results are depicted in Fig. 2. Ten studies (*Argyrou et al., 2020*; *Arjmandi et al., 2005*; *Braam et al., 2003*; *Chen et al., 1997*; *Dong et al., 2014*; *Grados et al., 2003*; *Macdonald et al., 2013*; *Nahas-Neto et al., 2018*; *Ooms et al., 1995*; *Stone et al., 2017*) reported using a randomized controlled design, but failed to detail their randomization method (*e.g.*, random number tables, computer randomization). Ten studies (*Aloia et al., 2013*; *Chen et al., 1997*; *Han et al., 2022*; *Holloway et al., 2007*; *Majidi et al., 2021*;

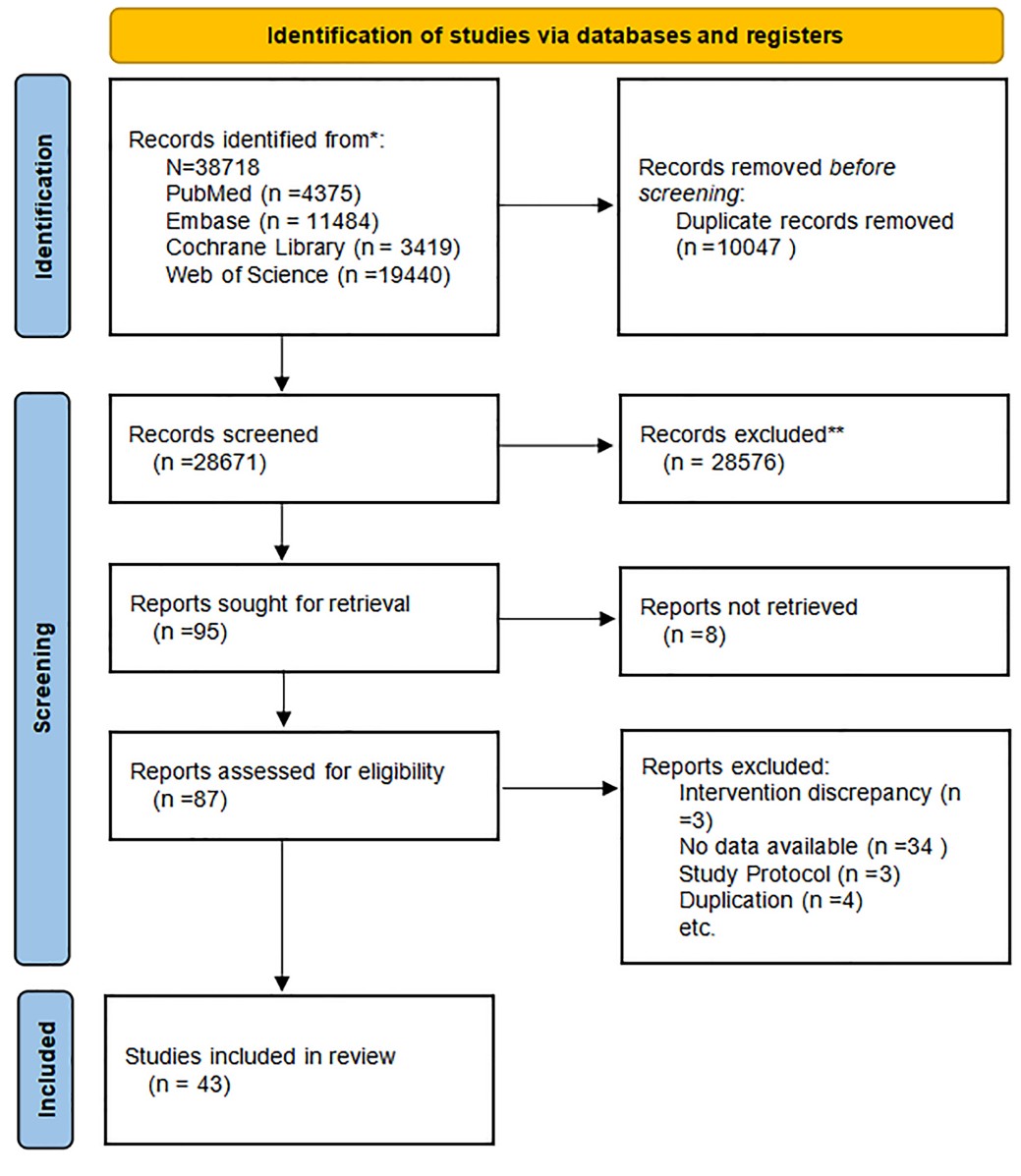

**Figure 1  Literature screening flowchart.**

*Pérez-Alonso et al., 2024*; *Prickett, Howe & Espiner, 2023*; *Ruml et al., 1999*; *Ulrich et al., 2004*; *Ye et al., 2006*) lacked adequate allocation concealment. One study (*Ulrich et al., 2004*) with missing outcome data faced a high bias risk, and two studies (*Atkinson et al., 2004*; *Ye et al., 2006*) showed a high risk of selective reporting bias. The overall risk of bias assessment revealed that 12 studies were at high risk, 11 studies were identified as having some concerns, and 20 studies were at low risk.

**Table 1  Basic information sheet.**

| First author | Year of publication | Study design | Region | Intervention (specific measures) | Number of cases | Age | Follow-up time | Outcome indicators |
|---|---|---|---|---|---|---|---|---|
| Ulrich | 2004 | cross-over design | USA | Calcium | 16 | 50.76 ± 2.65 | 18w | OC, BAP, ALP, NTX |
| | | | | Placebo | 11 | 52.91 ± 2.55 | 18w | |
| Meeta | 2022 | rct | India | Lycopene | 60 | 49.8 ± 3.9 | 6m | CTX, P1NP |
| | | | | Placebo | 48 | 49.7 ± 4.2 | 6m | |
| König | 2018 | rct | Germany | Protein | 66 | 63.8 ± 7.4 | 12m | CTX, P1NP |
| | | | | Placebo | 65 | 64.9 ± 7.1 | 12m | |
| Han | 2022 | rct | Korea | Probiotics | 27 | 58.4 ± 3.4 | 6m | OC, CTX |
| | | | | Placebo | 26 | 59.5 ± 3.4 | 6m | |
| Prickett | 2023 | cross-over design | New Zealand | Polyphenols | 60 | / | 12m | OC, CTX, ALP |
| | | | | Placebo | 65 | / | 12m | |
| Holloway | 2007 | cross-over design | Korea | Prebiotics | 15 | 72.2 ± 6.4 | 6w | OC |
| | | | | Placebo | 15 | 72.2 ± 6.4 | 6w | |
| Pérez-Alonso | 2024 | rct | Spain | Calcium + Vitamin D | 50 | 55 ± 3 | 12w | CTX, P1NP |
| | | | | Placebo | 49 | 55 ± 3 | 12w | |
| Ye | 2006 | rct | China | Polyphenols | 60 | 52.05 ± 3.11 | 6m | OC, BAP |
| | | | | Placebo | 30 | 52.7 ± 3.7 | 6m | |
| Vallibhakara | 2021 | rct | Thailand | Vitamin E | 26 | 63.15 ± 7.96 | 12w | CTX, P1NP |
| | | | | Placebo | 26 | 62.31 ± 6.50 | 12w | |
| Aloia | 2013 | rct | USA | Calcium + Vitamin D | 46 | 57.6 ± 7.1 | 28w | CTX, P1NP |
| | | | | Vitamin D | 47 | 59.7 ± 7.1 | 28w | |
| | | | | Calcium | 35 | 60 ± 8.5 | 28w | |
| | | | | Placebo | 31 | 58.6 ± 6.7 | 28w | |
| Ruml | 1999 | rct | USA | Calcium | 25 | 52.1 ± 4.1 | 24m | OC, BAP |
| | | | | Placebo | 31 | 51.7 ± 3.8 | 24m | |
| Panahande | 2019 | rct | Iran | Polyphenols | 21 | 57.38 ± 0.97 | 12w | CTX, P1NP, BAP |
| | | | | Placebo | 19 | 56.47 ± 3.64 | 12w | |
| Vanlint | 2012 | rct | Australia | Omega-3 | 20 | / | 12m | CTX |
| | | | | Placebo | 20 | / | 12m | |
| Koitaya | 2014 | rct | Japan | Vitamin K | 24 | 58.3 ± 4.0 | 12m | OC, BAP |
| | | | | Placebo | 24 | 58.5 ± 3.7 | 12m | |
| Jafarnejad | 2017 | rct | Iran | Probiotics | 20 | 58.85 ± 0.68 | 6m | OC, CTX, BAP, ALP |
| | | | | Placebo | 21 | 57.29 ± 0.72 | 6m | |
| Rajatanavin | 2013 | rct | Thailand | Calcium | 178 | 66.0 ± 4.4 | 24m | CTX, P1NP |
| | | | | Placebo | 165 | 65.6 ± 4.2 | 24m | |
| Mackinnon | 2011 | rct | Canada | Lycopene | 45 | 55.22 ± 1.04 | 4m | NTX |
| | | | | Placebo | 15 | 54.90 ± 0.72 | 4m | |

**Table 1** (*continued*)

| First author | Year of publication | Study design | Region | Intervention (specific measures) | Number of cases | Age | Follow-up time | Outcome indicators |
|---|---|---|---|---|---|---|---|---|
| Atkinson | 2004 | rct | UK | Polyphenols | 84 | 55.1 ± 4.72 | 12m | P1NP, ALP |
| | | | | Placebo | 90 | 55.2 ± 4.9 | 12m | |
| Majidi | 2021 | rct | Iran | Polyphenols | 22 | 55.57 ± 4.92 | 12w | OC, CTX |
| | | | | Placebo | 21 | 55.77 ± 3.65 | 12w | |
| Arjmandi | 2005 | rct | USA | Polyphenols | 35 | 53 ± 6 | 6m | OC, BAP, ALP |
| | | | | Placebo | 27 | 56 ± 5 | 6m | |
| Schult | 2004 | rct | USA | Polyphenols | 167 | 52.3 ± 2.99 | 12w | OC, NTX |
| | | | | Placebo | 85 | 52.3 ± 3.4 | 12w | |
| Herrmann | 2007 | rct | Australia | Vitamin B | 24 | 70 ± 6 | 12m | OC, CTX, P1NP |
| | | | | Placebo | 23 | 69 ± 6 | 12m | |
| Shen | 2018 | rct | USA | Vitamin E | 59 | 59.87 ± 7.03 | 12w | BAP, NTX |
| | | | | Placebo | 28 | 59.4 ± 6.3 | 12w | |
| Chen | 1997 | rct | Japan | Vitamin D | 25 | 52.82 ± 0.81 | 12m | OC, ALP |
| | | | | Calcium | 25 | 52.23 ± 0.67 | 12m | |
| Aoe | 2005 | rct | Japan | Protein | 14 | 50 ± 3 | 6m | OC, NTX |
| | | | | Placebo | 13 | 51 ± 3 | 6m | |
| Ooms | 1995 | rct | Netherlands | Vitamin D | 177 | 80.1 ± 5.6 | 24m | OC, BAP, ALP |
| | | | | Placebo | 171 | 80.6 ± 5.5 | 24m | |
| Bristow | 2014 | rct | New Zealand | Calcium | 77 | 71.47 ± 5.07 | 3m | CTX, P1NP |
| | | | | Placebo | 20 | 70 ± 3 | 3m | |
| Braam | 2003 | rct | Netherlands | Calcium + Vitamin D + Vitamin K | 56 | 55.3 ± 2.8 | 36m | OC, BAP |
| | | | | Calcium + Vitamin D | 46 | 55.7 ± 2.9 | 36m | |
| | | | | Placebo | 60 | 54.6 ± 2.8 | 36m | |
| Emaus | 2010 | rct | Norway | Vitamin K | 167 | 54.7 ± 2.5 | 12m | OC, BAP |
| | | | | Placebo | 167 | 54.2 ± 2.5 | 12m | |
| Tai | 2012 | rct | China | Polyphenols | 217 | 55.8 ± 3.6 | 96w | BAP, NTX |
| | | | | Placebo | 214 | 55.9 ± 4.0 | 96w | |
| Maria | 2017 | rct | USA | Vitamin D + Vitamin K | 11 | 60 ± 1.73 | 12m | OC, CTX, P1NP |
| | | | | Placebo | 11 | 57 ± 1.41 | 12m | |
| Koitaya | 2009 | rct | Japan | Vitamin K | 20 | 59.3 ± 3.7 | 4w | BAP |
| | | | | Placebo | 20 | 59.8 ± 3.1 | 4w | |
| Stone | 2017 | rct | USA | Vitamin B | 150 | 62.6 ± 8.7 | 7.3y | CTX, P1NP |
| | | | | Placebo | 150 | 62.5 ± 8.7 | 7.3y | |
| Vanitchanont | 2024 | rct | Thailand | Probiotics | 20 | 62 ± 5.07 | 12w | CTX, P1NP |
| | | | | Placebo | 20 | 64.05 ± 3.58 | 12w | |
| Macdonald | 2013 | rct | UK | Vitamin D | 174 | 64.56 ± 2.08 | 12m | CTX, P1NP |
| | | | | Placebo | 90 | 64.6 ± 2.3 | 12m | |
| Dong | 2014 | rct | USA | Omega-3 | 77 | 75 ± 6 | 6m | OC, BAP, NTX |
| | | | | Placebo | 39 | 75 ± 7 | 6m | |

**Table 1** (*continued*)

| First author | Year of publication | Study design | Region | Intervention (specific measures) | Number of cases | Age | Follow-up time | Outcome indicators |
|---|---|---|---|---|---|---|---|---|
| Takimoto | 2018 | rct | Japan | Probiotics | 31 | 57.5 ± 4.3 | 24w | BAP, NTX |
| | | | | Placebo | 30 | 57.8 ± 5.4 | 24w | |
| Cúneo | 2010 | rct | Argentina | Protein | 36 | 57.9 ± 4.8 | 24w | CTX, BAP |
| | | | | Placebo | 35 | 56.8 ± 4.8 | 24w | |
| Argyrou | 2020 | rct | Greece | Protein + Calcium + Vitamin D | 21 | 62.1 ± 6.3 | 3m | CTX, P1NP |
| | | | | Calcium + Vitamin D | 22 | 62 ± 7.6 | 3m | |
| Grados | 2003 | rct | France | Calcium + Vitamin D | 95 | 74.2 ± 6.4 | 12m | CTX |
| | | | | Placebo | 97 | 75.0 ± 7.3 | 12m | |
| Gregori | 2024 | rct | Sweden | Probiotics | 160 | 54.3 ± 2.99 | 24m | CTX, P1NP |
| | | | | Placebo | 79 | 54.65 ± 2.24 | 24m | |
| Nahas-Neto | 2018 | rct | Brazil | Vitamin D | 80 | 58.8 ± 6.6 | 9m | CTX, P1NP |
| | | | | Placebo | 80 | 59.3 ± 6.7 | 9m | |
| Wu | 2022 | rct | USA | Prebiotics | 10 | 63.1 ± 5.8 | 2m | CTX, P1NP |
| | | | | Placebo | 10 | 60.9 ± 4.7 | 2m | |

**Notes.**

rct, randomized controlled trial; y, year; m, month; w, week; CTX, C-terminal telopeptide of type I collagen; P1NP, Procollagen type I N-terminal propeptide; OC, Osteocalcin; BAP, Bone alkaline phosphatase; ALP, Alkaline phosphatase; NTX, N-terminal telopeptide of type I collagen.

*Aloia et al., 2013*; *Aoe et al., 2005*; *Argyrou et al., 2020*; *Arjmandi et al., 2005*; *Atkinson et al., 2004*; *Braam et al., 2003*; *Bristow et al., 2014*; *Chen et al., 1997*; *Cúneo et al., 2010*; *Dong et al., 2014*; *Grados et al., 2003*; *Gregori et al., 2024*; *Han et al., 2022*; *Herrmann et al., 2007*; *Holloway et al., 2007*; *Jafarnejad et al., 2017*; *Koitaya et al., 2009*; *Koitaya et al., 2014*; *König et al., 2018*; *Macdonald et al., 2013*; *Mackinnon et al., 2011*; *Majidi et al., 2021*; *Maria et al., 2017*; *Meeta et al., 2022*; *Emaus et al., 2010*; *Nahas-Neto et al., 2018*; *Ooms et al., 1995*; *Panahande et al., 2019*; *Pérez-Alonso et al., 2024*; *Prickett, Howe & Espiner, 2023*; *Rajatanavin et al., 2013*; *Ruml et al., 1999*; *Schult et al., 2004*; *Shen et al., 2018*; *Stone et al., 2017*; *Tai et al., 2012*; *Takimoto et al., 2018*; *Ulrich et al., 2004*; *Vallibhakara et al., 2021*; *Vanitchanont et al., 2024*; *Vanlint & Ried, 2012*; *Wu et al., 2022*; *Ye et al., 2006*.

## Results of the NMA
### Network diagram

In the NMA diagram, each node depicted an intervention. The node size reflected the number of studies involving the corresponding intervention. Larger nodes signified a greater number of included studies. Connections between nodes indicated direct comparisons of the two interventions. Line thickness represented the number of studies for each comparison; thicker lines meant more studies (Fig. 3). The node-splitting technique was utilized to examine inconsistency in each closed loop. The results revealed a *p*-value less than 0.05 for OC, indicating a discrepancy between direct and indirect comparisons. With respect to all other outcome measures, the *p*-values exceeded 0.05, indicating no statistically significant local inconsistency.

### CTX

Twenty-four studies in total provided data on CTX. The NMA revealed that, in postmenopausal women, calcium supplementation resulted in a statistically marked rise in CTX compared to placebo (calcium *vs.* placebo: MD = 0.11, 95% CI [0.03–0.18]), as illustrated in Fig. 4. The SUCRA probabilities ranked the interventions as follows: vitamin E (82.37%) > calcium (80.88%) > calcium + vitamin D + protein (70.57%). Vitamin E alone was associated with the greatest probability of achieving the largest reduction in CTX levels, detailed in Fig. 5.

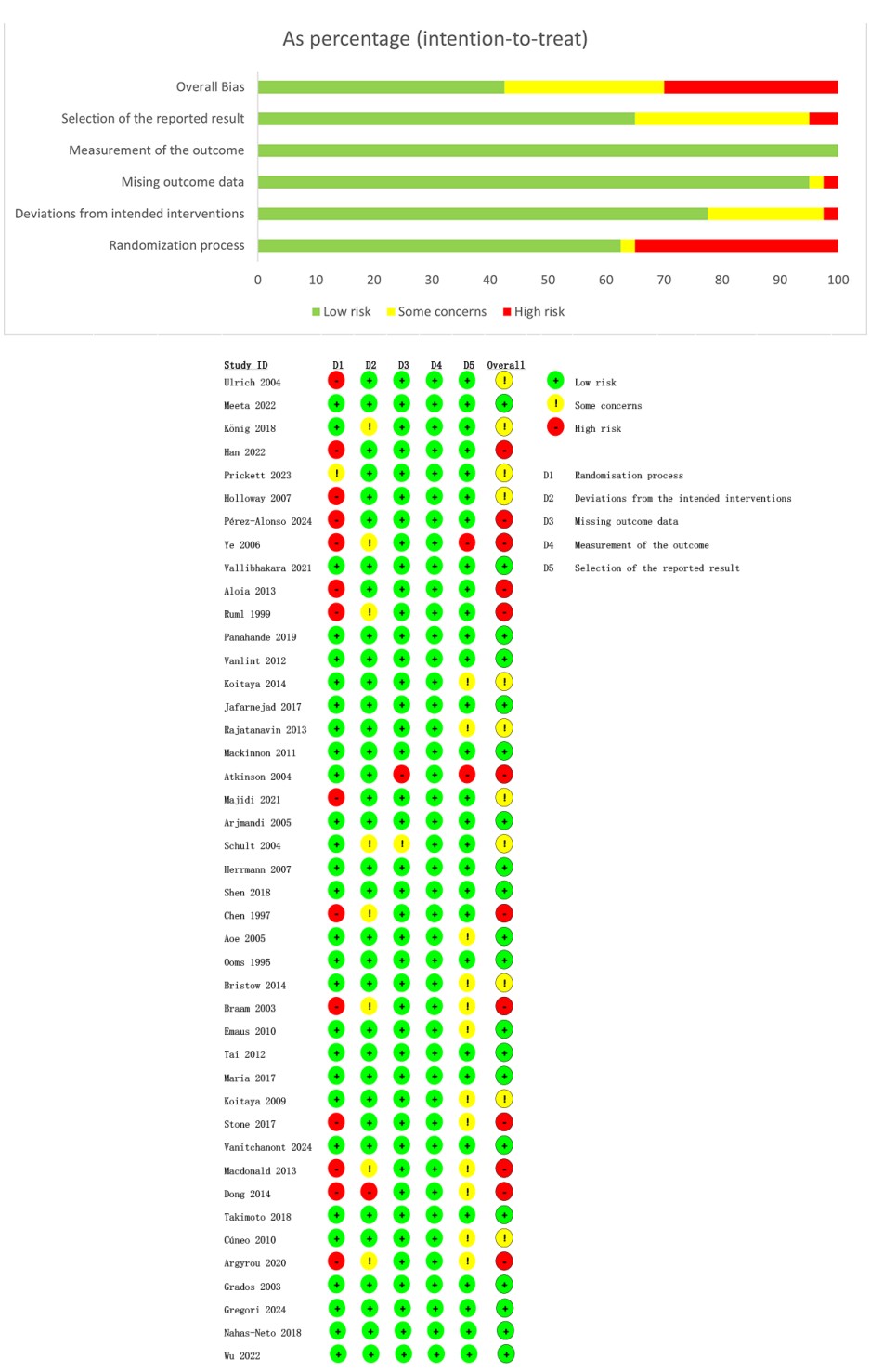

**Figure 2  Risk of bias evaluation.** Note: *Aloia et al., 2013*; *Aoe et al., 2005*; *Argyrou et al., 2020*; *Arjmandi et al., 2005*; *Atkinson et al., 2004*; *Braam et al., 2003*; *Bristow et al., 2014*; *Chen et al., 1997*; *Cúneo et al., 2010*; *Dong et al., 2014*; *Grados et al., 2003*; *Gregori et al., 2024*; *Han et al., 2022*; *Herrmann et al., 2007*; *Holloway et al., 2007*; *Jafarnejad et al., 2017*; *Koitaya et al., 2009*; *Koitaya et al., 2014*; *König et al., 2018*; *Macdonald et al., 2013*; *Mackinnon et al., 2011*; *Majidi et al., 2021*; *Maria et al., 2017*; *Meeta et al., 2022*; *Emaus et al., 2010*; *Nahas-Neto et al., 2018*; *Ooms et al., 1995*; *Panahande et al., 2019*; *Pérez-Alonso et al., 2024*; *Prickett, Howe & Espiner, 2023*; *Rajatanavin et al., 2013*; *Ruml et al., 1999*; *Schult et al., 2004*; *Shen et al., 2018*; *Stone et al., 2017*; *Tai et al., 2012*; *Takimoto et al., 2018*; *Ulrich et al., 2004*; *Vallibhakara et al., 2021*; *Vanitchanont et al., 2024*; *Vanlint & Ried, 2012*; *Wu et al., 2022*; *Ye et al., 2006*.

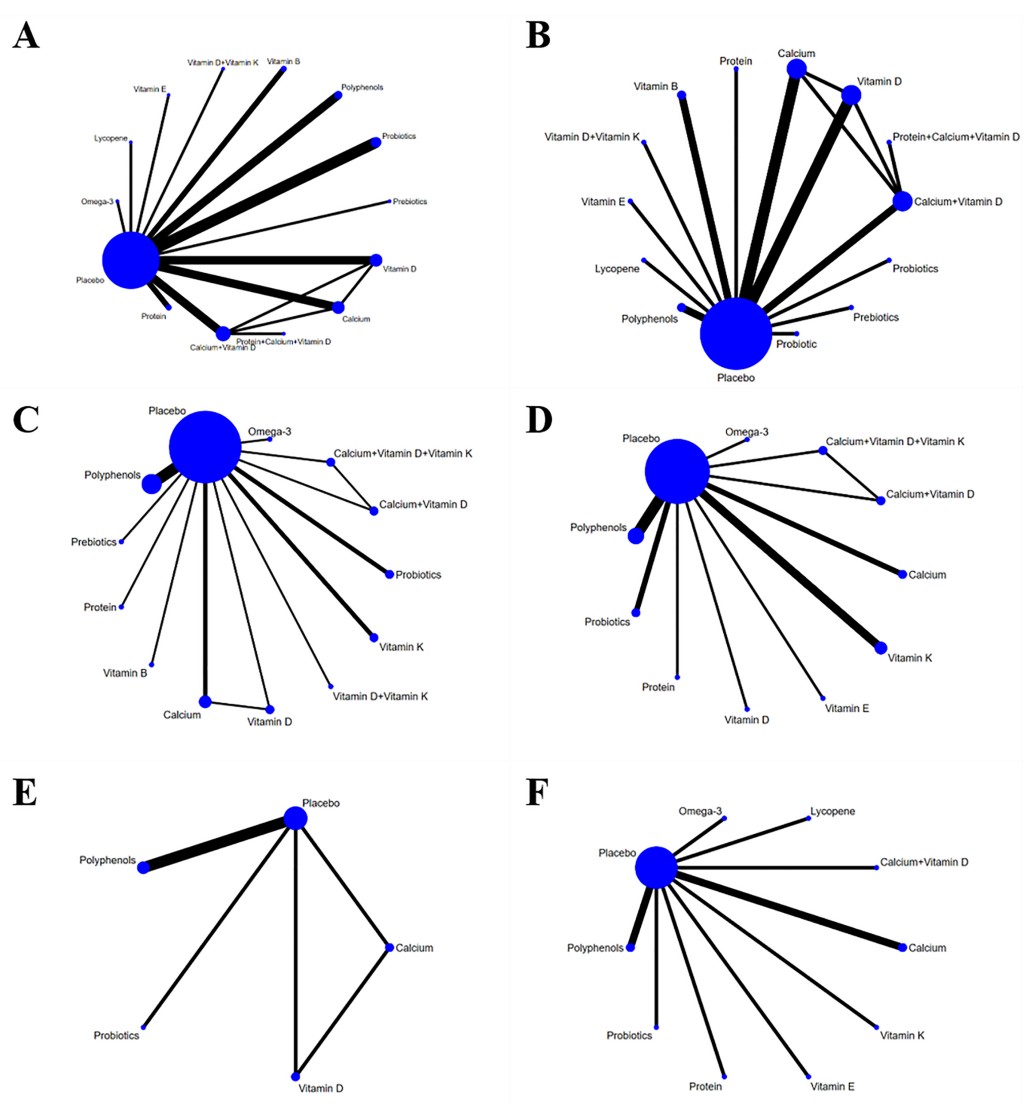

**Figure 3** Network diagram (A) CTX, (B) P1NP, (C) OC, (D) BAP, (E) ALP, (F) NTX.

Placebo
0.05 (-0.05, 0.16)  Protein
0.03 (-0.04, 0.1)  -0.02 (-0.15, 0.1)  Probiotics
-0.05 (-0.2, 0.1)  -0.1 (-0.29, 0.08)  -0.08 (-0.24, 0.08)  Prebiotics
0.01 (-0.06, 0.1)  -0.04 (-0.17, 0.1)  -0.02 (-0.12, 0.1)  0.06 (-0.1, 0.24)  Polyphenols
0.11 (0.03, 0.18)  0.05 (-0.08, 0.18)  0.08 (-0.03, 0.18)  0.16 (-0.01, 0.32)  0.09 (-0.02, 0.2)  Calcium
0.09 (-0.12, 0.3)  0.04 (-0.2, 0.27)  0.06 (-0.16, 0.28)  0.14 (-0.12, 0.4)  0.08 (-0.15, 0.3)  -0.02 (-0.24, 0.21)  Lycopene
-0.01 (-0.09, 0.09)  -0.06 (-0.2, 0.08)  -0.04 (-0.15, 0.08)  0.04 (-0.13, 0.22)  -0.02 (-0.15, 0.1)  -0.11 (-0.23, 0.01)  -0.1 (-0.32, 0.13)  Vitamin B
0.02 (-0.05, 0.09)  -0.03 (-0.16, 0.09)  -0.01 (-0.1, 0.09)  0.07 (-0.09, 0.24)  0.01 (-0.1, 0.11)  -0.08 (-0.18, 0.01)  -0.07 (-0.29, 0.15)  0.03 (-0.09, 0.14)  Vitamin D
0.12 (-0.01, 0.25)  0.07 (-0.1, 0.23)  0.09 (-0.05, 0.24)  0.17 (-0.02, 0.37)  0.11 (-0.05, 0.26)  0.02 (-0.13, 0.17)  0.03 (-0.21, 0.28)  0.13 (-0.03, 0.29)  0.1 (-0.05, 0.25)  Vitamin E
-0.02 (-0.18, 0.14)  -0.07 (-0.27, 0.12)  -0.05 (-0.22, 0.13)  0.03 (-0.18, 0.25)  -0.03 (-0.22, 0.14)  -0.12 (-0.3, 0.05)  -0.11 (-0.37, 0.15)  -0.01 (-0.19, 0.17)  -0.04 (-0.21, 0.13)  -0.14 (-0.34, 0.07)  Omega-3
0 (-0.8, 0.83)  -0.05 (-0.86, 0.78)  -0.03 (-0.84, 0.81)  0.05 (-0.77, 0.89)  -0.01 (-0.83, 0.82)  -0.1 (-0.91, 0.73)  -0.09 (-0.92, 0.77)  0.01 (-0.8, 0.84)  -0.02 (-0.83, 0.81)  -0.12 (-0.94, 0.72)  0.02 (-0.8, 0.87)  Vitamin D+Vitamin K
0.03 (-0.03, 0.14)  -0.02 (-0.14, 0.13)  0 (-0.09, 0.13)  0.09 (-0.07, 0.27)  0.02 (-0.08, 0.15)  -0.07 (-0.16, 0.05)  -0.05 (-0.27, 0.18)  0.04 (-0.07, 0.18)  0.01 (-0.07, 0.13)  -0.09 (-0.23, 0.08)  0.05 (-0.11, 0.25)  0.03 (-0.8, 0.85)  Calcium+Vitamin D
0.09 (-0.06, 0.27)  0.04 (-0.15, 0.25)  0.06 (-0.11, 0.26)  0.14 (-0.07, 0.38)  0.08 (-0.1, 0.27)  -0.02 (-0.18, 0.18)  0 (-0.26, 0.28)  0.1 (-0.08, 0.3)  0.07 (-0.1, 0.24)  -0.03 (-0.23, 0.19)  0.11 (-0.11, 0.35)  0.09 (-0.76, 0.91)  0.05 (-0.09, 0.2)  Calcium+Vitamin D+Protein

**Figure 4** Pairwise comparisons of effects of dietary supplements on reducing CTX.

## P1NP

Eighteen studies in total provided data on P1NP. Results from the NMA revealed that vitamin D and K resulted in a statistically marked elevation of P1NP in women after menopause compared to placebo, protein, probiotics, polyphenols, calcium, lycopene,

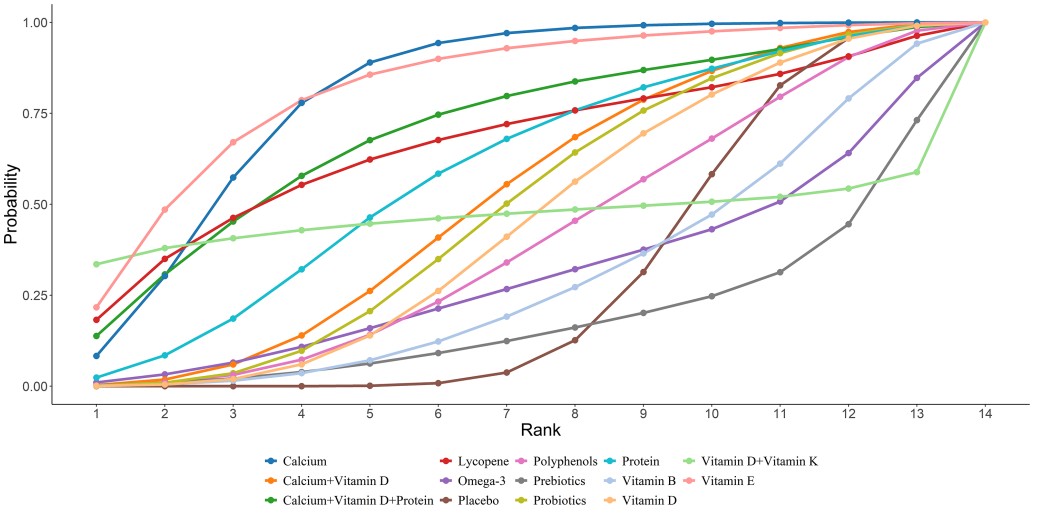

**Figure 5** SUCRA for the effect of different dietary supplements on reducing CTX.

**Figure 6** Pairwise comparisons of effects of dietary supplements on increasing P1NP.

vitamin B, vitamin D, vitamin E, calcium + vitamin D, and calcium + vitamin D + protein ((vitamin D + vitamin K *vs* placebo: MD = 36.27, 95% CI [29.34–43.17]; vitamin D + vitamin K *vs* protein: MD = 34.27, 95% CI [22.74–45.81]; vitamin D + vitamin K *vs* probiotics: MD = 25.36, 95% CI [17.61–33.05]; vitamin D + vitamin K *vs* prebiotics: MD = 36.08, 95% CI [21.18–51.09]; vitamin D + vitamin K *vs* polyphenols: MD = 29.57, 95% CI [19.32–39.81]; vitamin D + vitamin K *vs* calcium: MD = 45.81, 95% CI [38.1–53.55]; vitamin D + vitamin K *vs* lycopene: MD = 43, 95% CI [33.19–52.8]; vitamin D + vitamin K *vs* vitamin B: MD = 35.54, 95% CI= (27.05, 44); vitamin D + vitamin K *vs* vitamin D: MD = 39.36, 95% CI = (31.51, 47.19); vitamin D + vitamin K *vs* vitamin E: MD = 38.46, 95% CI = (27.28, 49.66); vitamin D + vitamin K *vs* calcium + vitamin D: MD = 39.27, 95% CI = (32.33, 46.21); vitamin D + vitamin K *vs* calcium + vitamin D + protein: MD = 45.45, 95% CI = (31.46, 59.54))). Further details are presented in Fig. 6. The SUCRA probabilities ranked the interventions as follows: vitamin D + vitamin K (100.00%) > probiotics (89.44%) > polyphenols (79.17%). Combined supplementation of vitamin D and vitamin K led to the greatest rise in P1NP levels (Fig. 7).

## OC

Nineteen studies in total provided data on OC. NMA results showed an obvious rise in OC levels in postmenopausal women taking vitamin D + vitamin K compared to placebo,
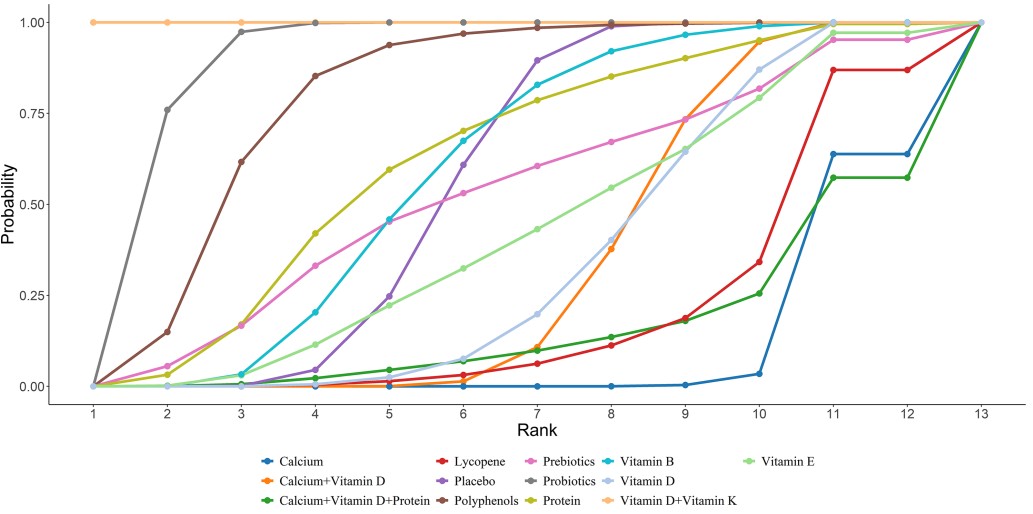

**Figure 7** SUCRA for the effect of different dietary supplements on increasing P1NP.

**Figure 8** Pairwise comparisons of effects of dietary supplements on increasing OC.

protein, polyphenols, calcium, vitamin B, vitamin D, vitamin K, Omega-3, calcium + vitamin D, and calcium + vitamin D + K ((vitamin D + vitamin K $vs$ placebo: MD = 5.88, 95% CI [2.17–9.6]; vitamin D + vitamin K $vs$ protein: MD = 5.68, 95% CI [0.74–10.62]; vitamin D + vitamin K $vs$ polyphenols: MD = 4.99, 95% CI [0.72–8.88]; vitamin D + vitamin K $vs$ calcium: MD = 6.92, 95% CI [2.56–11.1]; vitamin D + vitamin K $vs$ vitamin B: MD = 7.3, 95% CI [0.53–14.12]; vitamin D + vitamin K $vs$ vitamin D: MD = 4.86, 95% CI [0.33–9.26]; vitamin D + vitamin K $vs$ vitamin E: MD = 8.38, 95% CI [3.89–12.92]; vitamin D + vitamin K $vs$ calcium + vitamin D: MD = 6.28, 95% CI [1.12–11.49]; vitamin D + vitamin K $vs$ calcium + vitamin D: MD = 6.71, 95% CI [1.09–12.28]; vitamin D + vitamin K $vs$ calcium + vitamin D + vitamin K: MD = 7.09, 95% CI [1.55–12.63]). Further details are presented in Fig. 8. The SUCRA probabilities ranked the interventions as follows: vitamin D + vitamin K (97.05%) > Prebiotics (87.70%) > probiotics (79.15%). Combined vitamin D and K supplementation led to the greatest rise in OC levels (Fig. 9).

## BAP

Sixteen studies in total provided data on BAP. NMA results showed an obvious rise in BAP bone marker levels in postmenopausal women taking vitamin K compared to probiotics, calcium, and vitamin D [vitamin K $vs$ probiotics: MD = 4.35, 95% CI [3.14–5.56]; vitamin K $vs$ calcium: MD = 2.71, 95% CI [0.26–5.15]; vitamin K $vs$ vitamin D: MD =2.24, 95% CI

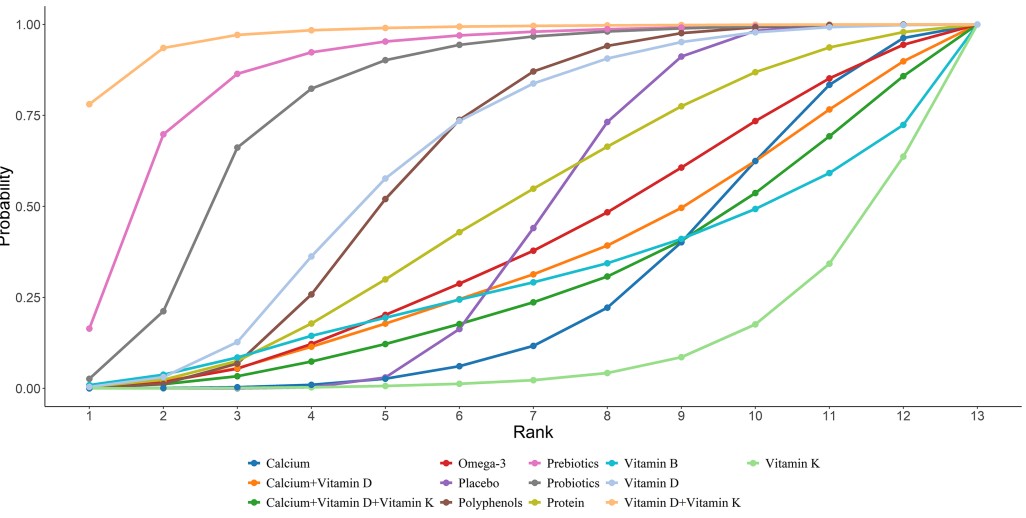

**Figure 9** SUCRA for the effect of different dietary supplements on increasing OC.

**Placebo**

| | | | | | | | | | | | | |
|---|---|---|---|---|---|---|---|---|---|---|---|---|
| 3.58 (-1.26, 8.43) | **Protein** | | | | | | | | | | | |
| **3.65 (2.84, 4.46)** | 0.07 (-4.86, 4.98) | **Probiotics** | | | | | | | | | | |
| 0.2 (-0.24, 0.65) | -3.38 (-8.24, 1.48) | **-3.45 (-4.37, -2.52)** | **Polyphenols** | | | | | | | | | |
| 2 (-0.26, 4.28) | -1.58 (-6.91, 3.76) | -1.64 (-4.06, 0.77) | 1.8 (-0.51, 4.11) | **Calcium** | | | | | | | | |
| 1.54 (-0.24, 3.33) | -2.04 (-7.19, 3.13) | **-2.1 (-4.06, -0.13)** | 1.34 (-0.5, 3.18) | -0.46 (-3.35, 2.43) | **Vitamin D** | | | | | | | |
| 3.23 (-0.74, 7.22) | -0.35 (-6.61, 5.91) | -0.42 (-4.48, 3.64) | 3.03 (-0.98, 7.03) | 1.22 (-3.36, 5.81) | 1.69 (-2.68, 6.04) | **Vitamin E** | | | | | | |
| -0.7 (-1.6, 0.19) | -4.28 (-9.2, 0.64) | **-4.35 (-5.56, -3.14)** | -0.91 (-1.91, 0.09) | **-2.71 (-5.15, -0.26)** | **-2.24 (-4.25, -0.24)** | -3.93 (-8.02, 0.14) | **Vitamin K** | | | | | |
| 1 (-2.4, 4.41) | -2.58 (-8.48, 3.33) | -2.64 (-6.15, 0.86) | 0.8 (-2.63, 4.25) | -1 (-5.07, 3.09) | -0.54 (-4.36, 3.3) | -2.22 (-7.47, 3.03) | 1.71 (-1.81, 5.23) | **Omega-3** | | | | |
| 1.1 (-0.83, 3.03) | -2.48 (-7.69, 2.74) | **-2.55 (-4.64, -0.45)** | 0.9 (-1.08, 2.88) | -0.9 (-3.89, 2.08) | -0.44 (-3.07, 2.19) | -2.12 (-6.54, 2.29) | 1.8 (-0.32, 3.94) | 0.09 (-3.82, 4.02) | **Calcium+Vitamin D** | | | |
| 1.4 (-0.59, 3.39) | -2.17 (-7.43, 3.06) | **-2.25 (-4.39, -0.1)** | 1.2 (-0.84, 3.24) | -0.6 (-3.62, 2.41) | -0.14 (-2.83, 2.53) | -1.83 (-6.27, 2.61) | 2.1 (-0.07, 4.29) | 0.39 (-3.55, 4.35) | 0.3 (-1.75, 2.35) | **Calcium+Vitamin D+Vitamin K** | | |

**Figure 10** Pairwise comparisons of effects of dietary supplements on increasing BAP.

[0.24–4.25]]. Further details are presented in Fig. 10. The SUCRA probabilities ranked the interventions as follows: vitamin K (95.50%) > placebo (81.61%) > polyphenols (72.91%). Vitamin K alone led to the greatest rise in BAP levels (Fig. 11).

### ALP

Seven studies in total provided data on ALP. NMA results showed an obvious rise in ALP bone marker levels in postmenopausal women taking calcium compared to placebo and vitamin D [calcium *vs* placebo: MD = 5.16 95% CI [0.53–9.78]; calcium *vs* vitamin D: MD = 6.2, 95% CI [2.95–9.45]]. Further details are presented in Fig. 12. The SUCRA probabilities ranked the interventions as follows: calcium (96.68%) > polyphenols (64.34%) > probiotics (37.41%). Calcium supplementation led to the greatest rise in ALP levels (Fig. 13).

### NTX

Eleven studies in total provided data on NTX. NMA results showed an obvious decrease in NTX bone marker levels in postmenopausal women taking protein compared to

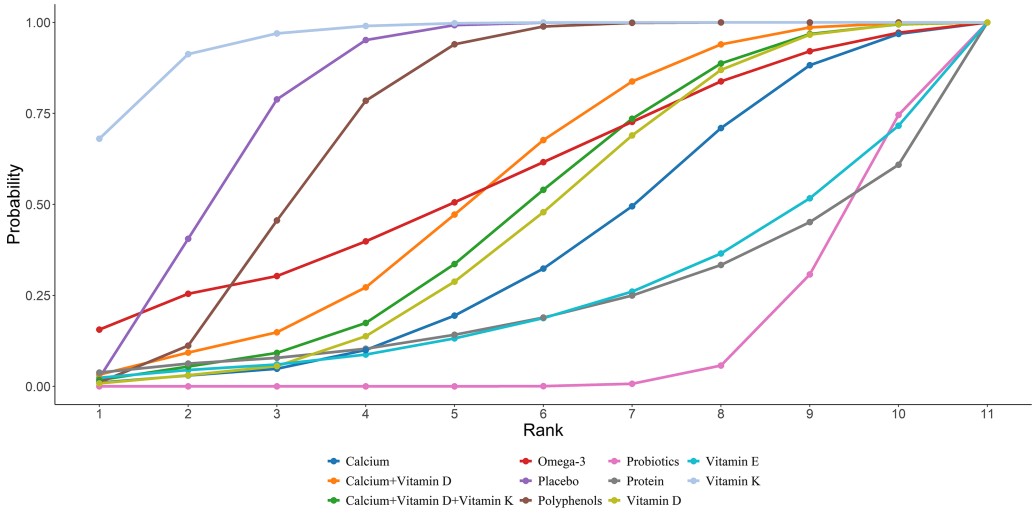

**Figure 11** SUCRA for the effect of different dietary supplements on increasing BAP.

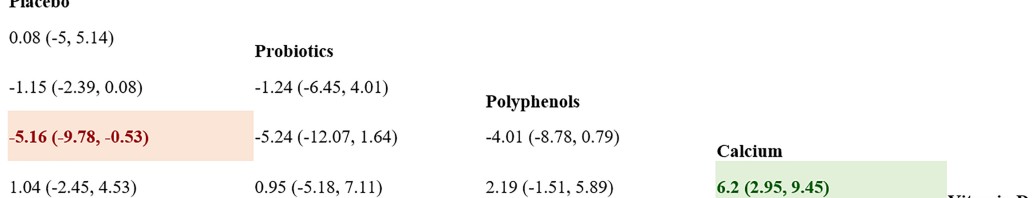

**Figure 12** Pairwise comparisons of effects of dietary supplements on increasing ALP.

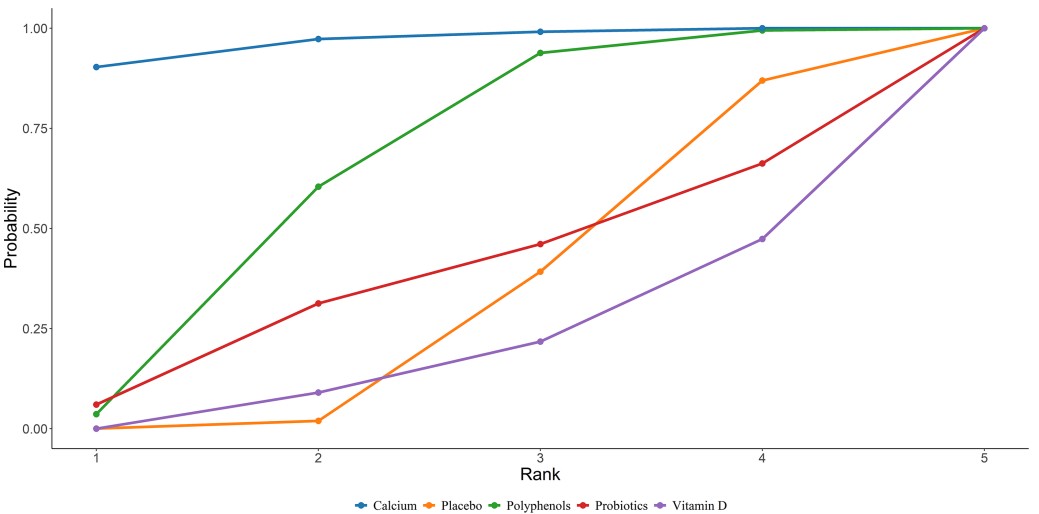

**Figure 13** SUCRA for the effect of different dietary supplements on increasing ALP.

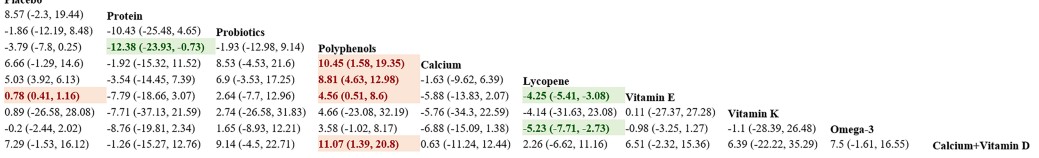

**Figure 14** Pairwise comparisons of effects of dietary supplements on reducing NTX.

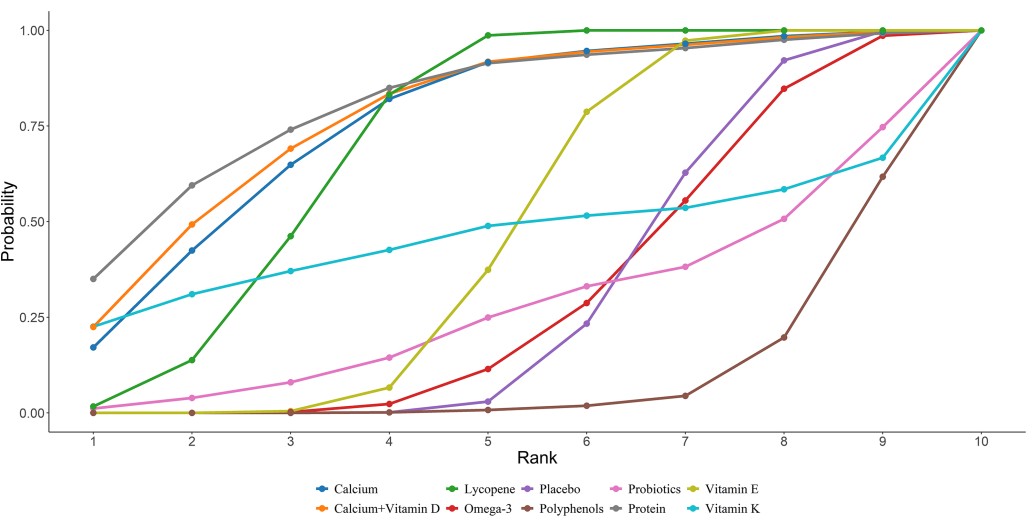

**Figure 15** SUCRA for the effect of different dietary supplements on reducing NTX.

polyphenols (protein *vs* polyphenols: MD $= -12.38$, 95% CI $[-23.93$ to $-0.73]$). Further details are presented in Fig. 14. The SUCRA probabilities ranked the interventions as follows: protein (81.20%) > calcium + vitamin D (78.27%) > calcium (76.38%). Calcium supplementation led to the greatest rise in ALP levels (Fig. 15).

### Publication bias

A comparison-correction funnel plot was applied to check for publication bias. The funnel plots were symmetrical, suggesting no significant publication bias (Fig. 16).

### DISCUSSION

To the best of our knowledge, this is the first NMA to examine the strength of dietary supplements on BTMs, including CTX, P1NP, OC, BAP, ALP, and NTX, in postmenopausal women. The current investigation included the latest data available from a total of 43 eligible trials. The findings revealed that, compared to other interventions, vitamin E supplementation alone resulted in the largest reduction in CTX. Combined supplementation of vitamin D and vitamin K significantly increased P1NP and OC levels. Vitamin K supplementation alone notably increased BAP. Calcium supplementation had

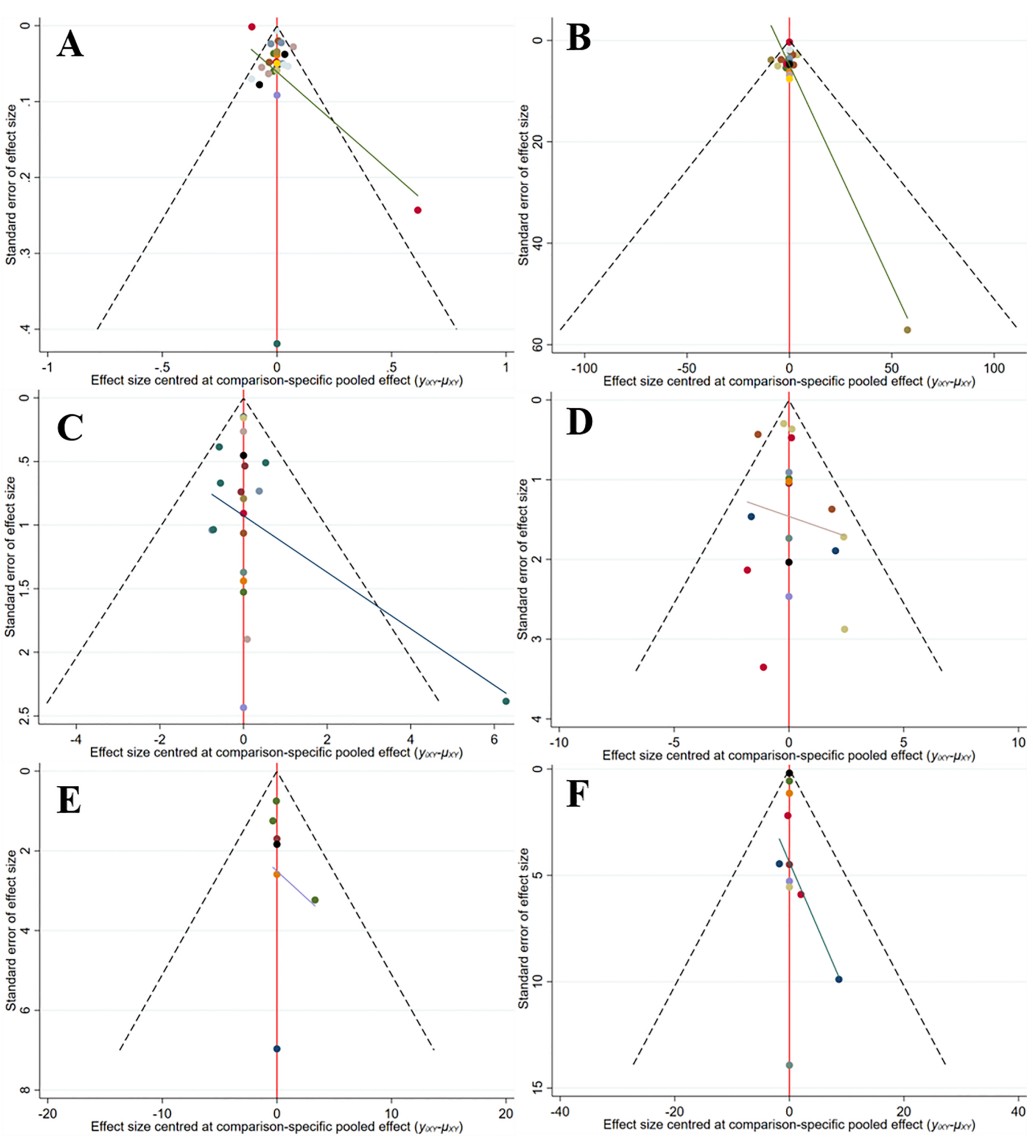

**Figure 16** Funnel plot (A) CTX, (B) P1NP, (C) OC, (D) BAP, (E) ALP, (F) NTX.

the greatest impact on increasing ALP, and protein intake was the most effective for reducing NTX.

BTMs are key indicators reflecting the processes of bone formation and resorption. They are products of enzymes secreted by osteoclasts during the degradation of bone collagen or the process of bone resorption. Bone formation markers are primarily synthesized and secreted by osteoblasts during various stages of bone formation. The serum levels of these markers could reflect the bone formation rate (*Vasikaran et al., 2024*). Common bone formation markers include P1NP, OC, BAP, and ALP, which are frequently used for monitoring bone diseases (*Anonymous, 2017*). Common bone resorption markers, including CTX and NTX, are released primarily by osteoclasts during bone resorption.

They are clinically used to assess bone resorption activity, particularly in the evaluation of osteoporosis, fracture healing, and the effectiveness of drug therapy (*Tian et al., 2019*).

Vitamin E supplementation alone was most effective in reducing CTX levels; however, no statistically significant difference was observed when compared to other interventions. *Vallibhakara et al. (2021)* observed that vitamin E alone or in combination with phytoestrogens exhibited a bone-protective effect on postmenopausal women with physiological bone loss. The treatment group demonstrated a trend of reduced serum CTX levels, whereas the placebo group showed a notable increase in serum CTX. Oxidative stress, a key factor in aging, exacerbates osteoporosis by increasing reactive oxygen species (ROS) levels (*Yu & Wang, 2022*). Elevated ROS promotes osteoclast activity, but does not affect osteoblast function, resulting in bone loss and thus accelerating osteoporosis. This process contributes to bone loss and osteoporosis among postmenopausal women and older adults (*Damani et al., 2022*; *Riegger et al., 2023*). Vitamin E, a strong antioxidant, mitigates reactive oxygen species (ROS) and several pro-inflammatory cytokines, including IL-1, IL-6, and TNF-α. These cytokines are essential for the activation of osteoclasts (*Chin & Ima-Nirwana, 2014*; *Domazetovic et al., 2017*; *Wong et al., 2019*). This demonstrates that vitamin E exerts an inhibitory effect on bone resorption during bone remodeling. *Shen et al. (2018)* also found that vitamin E suppressed bone resorption and serum soluble receptor activator of nuclear factor kappaB ligand (sRANKL), as well as elevated BALP/NTX ratio. The pathophysiology of postmenopausal bone loss is markedly influenced by antioxidant and anti-inflammatory effects of vitamin E. By suppressing the activation of nuclear factor-κB (NF-κB) and extracellular signal-regulated kinase (ERK) (*Chin, 2024*), vitamin E protects osteoblasts from lipid peroxidation, while simultaneously suppressing osteoclast differentiation, maturation, and bone resorption (*Ha et al., 2011*).

Vitamin K supplementation alone significantly increased BAP. Vitamin K (VK) is suggested to be a potent agent in mitigating bone loss in women, particularly those at perimenopausal or postmenopausal stages (*Aaseth et al., 2024*; *Kasukawa et al., 2014*). In obese animal models, vitamin MK-4 seems to have an osteogenic effect by upregulating activated OC and osteoprotegerin, and downregulating circulating RANKL (*Kim, Na & Sohn, 2013*). Activated osteocalcin seems to play a crucial role in directing calcium from the blood and other tissues into bone, where its binding to hydroxyapatite contributes to bone mineralization and reduces susceptibility to fractures (*Koitaya et al., 2009*). In addition, MK-4 has demonstrated effects on multiple genes and enzymes participating in osteogenesis, particularly bone-specific ALP, osteopontin (OPN), and osteoprotegerin (*Tabb et al., 2003*). Ultimately, *Alonso et al. (2023)* discovered that MK-7 could mitigate parathyroid hormone (PTH)-induced bone resorption.

Compared to other intervention measures, the simultaneous supplementation of vitamin D and vitamin K significantly elevated P1NP and OC. *Maria et al. (2017)* demonstrated that combined supplementation of vitamin D and vitamin K could increase the expression of P1NP and maintain stable levels of CTX, thereby effectively reducing bone turnover rate. Vitamin D supplementation increases the production of 1,25-dihydroxyvitamin D (1,25-$(OH)_2$D), a key factor promoting intestinal calcium absorption (*Park et al., 2024*; *Van Driel & Van Leeuwen, 2023*). This, in turn, reduces serum PTH levels by inhibiting

parathyroid function, thus decreasing bone turnover rate and consequently minimizing primarily cortical bone loss, ultimately leading to a significant reduction in the risk of fractures (*Gnoli et al., 2023*). Conversely, 1,25-$(OH)_2$D promotes the production of osteocalcin and alkaline phosphatase in osteoblasts, which may help counteract decreased bone turnover (*Emadzadeh et al., 2022*; *Hill & Aspray, 2017*). Consequently, vitamin D supplementation can enhance calcium utilization, promote the mineralization of previously under-mineralized bone, improve the rate of new bone mineralization, and reduce the incidence of fractures by decreasing bone turnover and bone loss. The findings from a meta-analysis encompassing 16 RCTs revealed that the combined intake of vitamin K2 and D3 improved BMD in postmenopausal women (*Ma et al., 2022*).

Among various interventions, calcium supplementation appears to be the most effective in increasing ALP levels. The findings from *Ulrich et al. (2004)* suggested that intake of at least 500 mg of calcium in healthy postmenopausal old women moderately elevated lumbar spine BMD, attenuated bone loss in the femoral neck region, and reduced BTMs. OC levels rose markedly, contrasting with a decline in P1NP. According to *McKane et al. (1996)*, if old women fail to consume sufficient calcium through their diet to meet threshold needs, this may result in enhanced parathyroid hormone activity and accelerated bone resorption, ultimately impacting bone metabolic equilibrium (*Akesson et al., 1998*). This involves imbalances in calcium homeostasis, dysregulation of hormonal control, and disruption of the dynamic equilibrium of bone metabolism (*Reginster et al., 1998*). Consequently, sufficient calcium supplementation can contribute to improved BMD and decreased bone loss among postmenopausal women, which is vital for the prevention of osteoporosis.

Our results indicated that protein consumption was the most efficacious in reducing NTX levels compared to other interventions. Over a 12-month period, specific collagen peptides (SCP) supplementation led to a notable rise in procollagen type 1 N-terminal propeptide (P1NP), a marker of bone formation, whereas the control group showed a substantial increase in collagen type I C-telopeptide (CTX), a bone resorption marker (*König et al., 2018*). This result indicates that by both encouraging bone formation and hindering bone resorption, SCP appears to regulate bone metabolism, potentially improving bone health (*Ahn & Je, 2019*). Preclinical *in vitro* studies or rodent model studies indicate that supplementation of collagen peptides significantly increases bone organic components, improves bone metabolism and microstructure, and enhances vertebral biomechanical resistance (*Bu et al., 2021*). The precise mechanisms by which collagen peptides promote bone formation and increase BMD remain unclear. However, studies suggest that collagen peptides are quickly taken up by the body from the digestive system. These peptides may then act as signaling molecules, which can have a positive impact on anabolic processes (*Oesser & Seifert, 2003*; *Walrand et al., 2008*). Research on rodents further demonstrates that collagen peptides notably elevate the levels of bone organic matrix, which is closely associated with a rise in BMD (*Watanabe-Kamiyama et al., 2010*).

In this study, we thoroughly searched related literature to comprehensively collect evidence currently available concerning supplement use among postmenopausal women. Certain limitations exist in this study. First, while we included all RCTs with available data,

a lack of trials allowing for direct comparisons may have influenced our results. Secondly, the included RCTs varied in their designs, populations, and durations, and also utilized various outcome indicators. Given the limitations of the quality and consistency of the data included, it is challenging to draw reliable conclusions. Thirdly, some included studies failed to report on patient blinding or allocation concealment, which could adversely affect the overall assessment of study quality. Fourth, the non-significant or inconsistent effects of certain supplements on some BTMs may stem from variations in baseline characteristics of study participants, differences in intervention dosages and durations, or heterogeneity in study designs, leading to discrepancies in research findings. Lastly, no subgroup analysis is performed, and thus it is infeasible to determine whether variations exist among different populations or geographical regions, thus precluding the formulation of the most appropriate treatment recommendations for diverse patient groups.

In conclusion, dietary supplements may serve as an important strategy for regulating bone metabolism and enhancing bone health in women after menopause. By actively promoting bone formation and preventing bone resorption, vitamin D, vitamin E, vitamin K, calcium, and protein can contribute significantly to managing the progression of osteoporosis. As the first NMA comprehensively comparing the effects of various dietary supplements on BTMs, this study addresses the limitations of previous single-nutrient investigations or conventional meta-analyses. It provides a more comprehensive chain of evidence for investigating the mechanisms of bone metabolism regulation, elucidates the multifaceted regulatory network of nutrient intervention on bone metabolism, and offers a theoretical basis for targeted intervention strategies. Given that head-to-head comparisons are frequently absent in RCTs investigating dietary supplements, the data derived from these trials should be carefully interpreted. Further RCTs are needed to investigate the effects of dietary supplements on BTMs, along with the underlying mechanisms.

### Funding
The authors received no funding for this work.

### Competing Interests
The authors declare there are no competing interests.

### Author Contributions
- Yan Wei conceived and designed the experiments, analyzed the data, authored or reviewed drafts of the article, and approved the final draft.
- Congjie Lei conceived and designed the experiments, performed the experiments, authored or reviewed drafts of the article, and approved the final draft.
- Yue Zhong performed the experiments, authored or reviewed drafts of the article, and approved the final draft.
- Hongchun Shen conceived and designed the experiments, performed the experiments, analyzed the data, prepared figures and/or tables, authored or reviewed drafts of the article, and approved the final draft.

## Data Availability

No new data were created or analyzed in this systematic analysis/meta-analysis.

## Supplemental Information

Supplemental information for this article can be found online at http://dx.doi.org/10.7717/peerj.19882#supplemental-information.

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
