# Peer review of "Effects of dietary supplements on bone turnover markers in women after menopause: a network meta-analysis"

_PeerJ, doi:10.7717/peerj.19882_

## Round 0.1 · original submission · Major Revisions

Reviewer 1 ·

Basic reporting

The Introduction should clearly explain the rationale for selecting the specific dietary supplements and bone turnover markers (BTMs) evaluated in the study. It is important to elaborate on how each supplement influences bone metabolism, detailing the underlying biological mechanisms by which these supplements affect the selected BTMs.

Experimental design

Use of a Bayesian Network Meta-Analysis (NMA) is appropriate for comparing multiple interventions and offers a strong quantitative approach. Comprehensive Database Search including PubMed, Embase, Cochrane, and Web of Science ensures broad coverage. Employing ROB 2.0 enhances the credibility of the included trials.

Validity of the findings

The study’s conclusions are clearly articulated, directly address the original research question, and are appropriately grounded in the supporting results. However, for findings that are insignificant or contradictory, it would be helpful to provide possible explanations or hypothesised mechanisms. Offering some interpretation can help readers better understand the context and potential implications of these results.

Additional comments

Please refer to the attached pdf with comments.

Annotated reviews are not available for download in order to protect the identity of reviewers who chose to remain anonymous.

Reviewer 2 ·

Basic reporting

The content of the introduction is interesting and relevant.
I recommend reviewing and revising all references to ensure accurate reporting.
References: Please use original references. For example, you cite "Its global prevalence is approximately 18.3%, with about 30% of postmenopausal women affected (Yang et al. 2024)." However, the Yang et al paper is a systematic review and meta-analysis of the effects of vitamin D and bisphosphonates and not the prevalence of osteoporosis. Or you cite Ko & Kim 2020 in this sentence: "These changes elevate the likelihood of osteoporosis and fractures". However Ko and Kim's paper in on menopause lipid associated disorders and foods beneficial for postmenopausal women.

Please provide a brief and non-technical explanation for readers about what “a network meta-analysis (NMA) methodology” is in the introduction section before providing rationale for using it.
Tables: Please include other abbreviations in the legends of the table for (rct, y, w and m)
Figures: The font of the pairwise comparisons is too small on all those figures; please use a landscape page layout instead of portrait. For example, it is not possible to read Figure 4’s numbers. Also, figures have no descriptive captions. Please add captions to all figures.

Experimental design

Well conducted and rigorous study with high technical and ethical standards.. The research question is well-defined, relevant to clinicians and nutritionists, and might have a meaningful impact on patients and helps fill a knowledge gap in the field of nutrition and bone health.

Statistical Analysis section :

Please clarify what “the annealing value of 10,000” in the context of MCMC is on line 143.

On line 148 by “An overall I2 statistic of less than 50% was interpreted as acceptable
149 heterogeneity and supportive of the assumption of homogeneity.” Do you mean that an I² value of less than 50% indicates low to moderate heterogeneity? Please clarify.

You mention “Convergence of patient demographics and study outcomes was evaluated through a quality assurance test.” Please clarify what specific metrics or methods were used for this evaluation.

Please describe how you interpreted forest and funnel plots, either in the figure captions or in the statistics section.

Validity of the findings

Aside from comments above, results appear to be robust, statistically sound, & controlled.

Please provide some evidence on the clinical relevance and relationship between bone markers, bone density, and risk of fractures in the rationale section of the paper in the latter part of the introduction section. It would be helpful to know for readers what the clinical value is of improving bone marker profiles in preventing osteoporotic fractures, and if there is any clinical evidence that altering bone markers in postmenopausal women would benefit their bone density and/or risk of fractures.

This paper may be of great interest to clinicians, and for that reason, it would be helpful to make it more reader-friendly by adding captions to figures, explaining in the stats section why some of the methods are used, and the clinical implications of the findings in the discussion section.

Conclusions are well stated, linked to the original research question & limited to supporting results

Additional comments

This is a well-written paper with rigorous statistical methodology and may be of great interest to clinicians and the general public. The authors have put in a great deal of work and effort. It would be helpful to add more descriptions and explanations to the figure and tables, and to address the comments above.

---

## Round 0.2 · accepted · Accept

The authors have addressed all of the reviewers' comments and manuscript is ready for publication.

Reviewer 1 ·

Basic reporting

-

Experimental design

-

Validity of the findings

-

Additional comments

The authors have addressed all my comments/suggestions.

Reviewer 2 ·

Basic reporting

All issues raised by me and the other reviewer were adequately addressed by the authors.

Experimental design

-

Validity of the findings

-